# Serum Proteomic Profile of Asthmatic Patients after Six Months of Benralizumab and Mepolizumab Treatment

**DOI:** 10.3390/biomedicines10040761

**Published:** 2022-03-24

**Authors:** Lorenza Vantaggiato, Paolo Cameli, Laura Bergantini, Miriana d’Alessandro, Enxhi Shaba, Alfonso Carleo, Fabrizio Di Giuseppe, Stefania Angelucci, Guido Sebastiani, Francesco Dotta, Luca Bini, Elena Bargagli, Claudia Landi

**Affiliations:** 1Functional Proteomics Lab, Department of Life Sciences, University of Siena, 53100 Siena, Italy; lorenz.vantaggiato@student.unisi.it (L.V.); enxhi.shaba@unisi.it (E.S.); luca.bini@unisi.it (L.B.); 2Complex Operational Unit of Respiratory Diseases and Lung Transplantation, Department Internal and Specialist Medicine, University of Siena, 53100 Siena, Italy; paolo.cameli@unisi.it (P.C.); laura.bergantini@unisi.it (L.B.); miriana.dalessand@student.unisi.it (M.d.); 3Department of Pulmonology, Hannover Medical School, 30625 Hannover, Germany; carleo.alfonso@mh-hannover.de; 4Proteomics Unit, Department of Medical, Oral and Biotechnological Sciences, University “G. d’Annunzio” Chieti-Pescara, 66100 Chieti, Italy; f.digiuseppe@unich.it (F.D.G.); s.angelucci@unich.it (S.A.); 5Diabetes Unit, Department of Medical Sciences Surgery and Neurosciences, University of Siena, 53100 Siena, Italy; guido.sebastiani@unisi.it (G.S.); francesco.dotta@unisi.it (F.D.)

**Keywords:** mepolizumab, benralizumab, ceruloplasmin, plasminogen, ApoA1, ApoC, severe eosinophilic asthma

## Abstract

Severe eosinophilic asthma is characterized by chronic airway inflammation, oxidative stress, and elevated proinflammatory cytokines, especially IL-5. Mepolizumab and benralizumab are both humanized IgG antibodies directed against IL-5 signaling, directly acting on eosinophils count. Together with the complexity of severe asthma classification and patient selection for the targeted treatment, there is also the urgency to clarify the follow-up of therapy to identify biomarkers, in addition to eosinophils, for the optimal duration of treatment, persistence of effectiveness, and safety. To this purpose, here we performed a follow-up study using differential proteomic analysis on serum samples after 1 and 6 months of both therapies and sera from healthy patients. Statistical analysis by PCA and heatmap analyses were performed, and identified proteins were used for enrichment analysis by MetaCore software. The analysis highlighted 82 differences among all considered conditions. In particular, 30 referred to benralizumab time point (T0, T1B, T6B) and 24 to mepolizumab time point (T0, T1M, T6M) analyses. t-SNE and heatmap analyses evidence that the differential serum protein profile at 6 months of both treatments is more similar to that of the healthy subjects. Among the identified proteins, APOAI, APOC-II, and APOC-III are upregulated principally after 6 months of benralizumab treatment, plasminogen is upregulated after 6 months of both treatments and ceruloplasmin, upregulated already after 1 month of benralizumab, becoming higher after 6 months of mepolizumab. Using enrichment analysis, identified proteins were related to lipid metabolism and transport, blood coagulation, and ECM remodeling.

## 1. Introduction

Severe eosinophilic asthma (SEA) represents one of the major endotypes of severe asthma, usually characterized by chronic inflammation of airways, with recurrent exacerbation and sputum and peripheral blood hypereosinophilia [1]. Moreover, structural changes due to the increase in reticular sub-basement membrane by direct interaction of eosinophils with fibroblasts, are responsible for variable expiratory airflow limitation and bronchial remodeling [2]. Activated eosinophils secrete granules containing major basic protein (MBP) and eosinophil peroxidase (EPO) [3,4] that induced bronchial epithelial cells to release many growth factors [5]: TGF-α, TGF-β1, PDGF-β, EGFR, and metalloproteinase (MMP). This protein release alters bronchial epithelium integrity and function and induces extracellular matrix (ECM) components’ deposition [6,7]. Eosinophils’ proliferation, maturation, and activation are dependent on IL-5 concentrations, mainly produced by Th2 cells, group 2 innate lymphoid cells (ILC2), mast cells, natural killer T (NKT) cells, and eosinophils themselves [8]. To this purpose, different therapies against the IL-5 pathway have been developed and approved for SEA control; among them are mepolizumab and benralizumab. Both are murine humanized monoclonal N-glycosylated IgG1/k antibodies (MAb) and mainly target T2-high airway inflammation that clinically manifested a combination of peripheral eosinophilia, sputum eosinophilia, and/or elevated fractional exhaled nitric oxide (FENO) associated to recurrent exacerbation of disease requiring oral steroids [9,10]. While mepolizumab binds IL-5 α-chain and blocks the activity of circulating IL-5, benralizumab targets the receptor of IL5, in particular, the subunit IL-5Rα, mainly expressed in eosinophils. Furthermore, benralizumab has high affinity also for human FcγRIIIa expressed on natural killer cells (NK), causing aggregation around the eosinophil and resulting in antibody-directed cell-mediated cytotoxicity and eosinophil apoptosis [8].

Different studies reported that already after 1 month of both treatments, patients reported a significant symptomatic improvement and reduction of the need for rescue therapy with SABA or ICS/LABA and improved significantly in terms of functional and immunological parameters, as well as clinical status [9,11,12,13]. The treatment with the two MAbs also determines a reduction in ECM protein deposition in the sub-basement membrane improving the expiratory airflow [14]. Unfortunately, these target treatments cannot be considered as the “panacea”, as a significant proportion of patients do not respond completely. Accordingly, together with the complexity of severe asthma classification and patient selection for the targeted treatment, there is also the urgency to clarify the follow-up of therapy to identify biomarkers for the optimal duration of treatment, persistence of effectiveness, and safety. To this purpose, we applied a differential proteomic approach to the SEA patient serum in order to highlight differential proteins over the therapy follow-up to further confirm our previous results. Our previous research reported differential serum proteomic profiles after 1 month of both therapies. In particular, several protein species of ceruloplasmin increase in abundance after 1 month of benralizumab compared to after mepolizumab treatment [12,13]. Moreover, applying a redox proteomic approach, we also realized that ceruloplasmin was also highly oxidized after benralizumab compared to mepolizumab [15]. Therefore, these preliminary works lead us to carry on with further analyses for a longer period of follow-up. Here, we investigate, by differential proteomic approach, the drug-induced molecular modifications comparing serum proteomic profiles of patients with SEA, before and after 1 and 6 months of mepolizumab and benralizumab, also taking into consideration healthy controls.

## 2. Materials and Methods

### 2.1. Experimental Design

Eighteen patients with severe eosinophilic asthma that reported blood eosinophil count ≥300 cells/μL were treated with monoclonal therapies. Ten patients were treated with mepolizumab and eight with benralizumab. The serum samples were collected at three different times: pre-treatment (T0), after 1 month of treatment with benralizumab (T1B) or mepolizumab (T1M), and after 6 months of treatment with benralizumab (T6B) or mepolizumab (T6M). The cohort of patients was in part used for proteomic analysis and in part used for validation by WB. Samples used for proteomic analysis were randomly selected.

Differential proteomic analysis by two-dimensional electrophoresis was performed on 5 patients treated with mepolizumab and 4 patients treated with benralizumab at different time points, also taking into consideration healthy controls. Spots that resulted statistically significant from the comparison were identified by mass spectrometry MALDI-TOF. Identified proteins were submitted to enrichment analysis in order to highlight protein position in specific molecular pathways. Proteins with central functional roles and their behavior were validated by mono and bidimensional Western blot in an alternative cohort of samples to confirm their particular pattern and their amount in precise time points of benralizumab and mepolizumab treatments. Proteomic data of the differential proteins were also correlated with peripheral eosinophilia (cell/mm^3^%) by Pearson’s linear correlation.

### 2.2. Population

For this study, a cohort of 18 adult Caucasian patients suffering from severe eosinophilic asthma, treated with anti-IL-5 biological therapies, was recruited. Eight patients were treated with benralizumab and 10 with mepolizumab. The pathology was diagnosed according to international guidelines, as previously reported [11]. Treatment doses were for a subcutaneous administration of 100 mg every 4 weeks mepolizumab and a subcutaneous administration of 30 mg every 4 weeks in the first 3 months and every 8 weeks subsequently for benralizumab. Seven healthy adults were included in the analysis as controls and the inclusion criteria were: normal values of lung function parameters, not suffering from asthma or other chronic or infection diseases, not on treatment for any illness. Clinical, functional, and immunological data were monitored at Siena University Regional Referral Centre for Rare Lung Diseases and were entered in an electronic database. All patients provided written informed consent for participation in the study. The Local Ethics Committee approved the study (OSS_REOS code number 12908).

### 2.3. Serum Preparation for 2DE Analysis

Blood samples were drawn in the morning, after 8 h of fasting, into serum tubes (BD vacutainer, SST II Advance, Plymouth, UK). Samples were centrifuged for 10 min at 1690× *g*, then serum samples were recovered and stored at −80 °C until analysis. Sample preparation for proteomic analysis was executed as previously described [12]; briefly 10 μL serum was diluted in 16 μL of denaturing buffer (and 2.3% DTE *w/v* and 10% SDS *w/v*) and then warmed to 95 °C for 7 min. When cooled at room temperature, conventional lysis buffer solution (8 M UREA *w/v*, 4% CHAPS *w/v*, 1% DTE *w/v*, and bromophenol blue in trace) was added to reach a final volume of 500 μL. For the analytical run, 60 μg proteins, contained in 50 μL of sample prepared in this way, with 0.2% (*v/v*) carrier ampholyte was loaded by cup-loading on IPFstrips (Cytiva, formerly GE Healthcare), rather 120 μg proteins with 0.2% (*v/v*) carrier ampholyte was cup-loaded for 2D Western blot (2DWB). For the mass spectrometry, a preparatory run of 10 μL of serum sample was denaturized in 90 μL of lysis buffer solution, obtaining 600 μg of proteins to which 2% (*v/v*) carrier ampholyte was added and that was cup-loaded. 

### 2.4. Two-Dimensional Electrophoresis

For Two-Dimensional Gel Electrophoresis (2-DE), isoelectric focusing (IEF) was carried out on precast 18 cm-long non-linear Immobiline Dry-Strip, with a range of immobilized pH gradient 3–10 (Cytiva, formerly GE Healthcare, Chicago, IL, USA) and Ettan™ IPGphor™ system (Cytiva). IPGstrips were rehydrated with 350 μL of lysis buffer containing bromophenol blue in trace for 12 h at room temperature. For both the analytical runs and MS-preparatory runs, proteins were focused at 16 °C, according to the following voltage program: 200 V for 8 h, a gradient to 3500 V in 2 h, a step at 3500 V for 2 h, from 3500 to 5000 V in 2 h, maintained at 5000 V for another 3 h, a gradient to 8000 V in 1 h, and a step at 8000 V for 3 h. Finally, a gradient to 10,000 V in 1 h and maintained up to a total of 100,000 Vh.

After IEF, the IPGstrips underwent an equilibration step according to Landi et al. [15]. The second-dimension run was performed on 9–16% polyacrylamide linear gradient gels, and carried out at 40 mA/gel constant current at 9 °C. For analytical and transfer gels, ammoniacal silver staining was used [16], while MS-preparative gels were stained according to Rabilloud MS-compatible silver staining protocol [17]. All gels were digitalized using the Image Scanner III coupled with the LabScan 6.0 software (Cytiva, formerly GE Healthcare).

### 2.5. Image Analysis

Gel image analysis for spot comparison was carried out using Image Master 2D Platinum 6.0 software (Cytiva, formerly GE Healthcare). In order to find quantitative and qualitative differences, an intra-class analysis was performed matching all gels from the same condition with their Master reference gels. Then, Masters were matched together in inter-class analysis, using the relative volume (%V) (integration of optical density of a single spot (volume) divided by the total volume of spots and expressed as a percentage) as the evaluation parameter. We considered differentially abundant spots as those having a difference in %V mean ratio by at least ±2-fold between T0, T1M, T1B, T6M, T6B, and CTRL, and validated by Kruskal–Wallis test (*p* ≤ 0.05) followed by comparisons of mean ranks by Dunn’s test. Statistical analysis was performed using RStudio Desktop 1.1.463 (Integrated Development for RStudio, Inc., Boston, MA, USA, www.rstudio.com; 3 May 2021).

### 2.6. Monodimensional and Two-Dimensional Western Blot

For 1DWb, 10 μL from four serum samples for each condition, prepared as previously described, were combined with Laemmli buffer: 2% (*w/v*) SDS, 100 mM Tris–HCl pH 6.8, 4% (*v/v*) β-mercaptoethanol, 20% (*v/v*) glycerol, and boiled at 95 °C for 5 min. For each sample 20 μg of proteins were loaded and resolved on 10% polyacrylamide gel and afterward transferred onto a nitrocellulose membrane (Hybond ECL, Cytiva). For 2DWb we ran a sample for each condition. After 2DE separation, gels were equilibrated in Towbin transfer buffer (25 mM Tris, 192 mM glycine, 20% *v/v* methanol) for 1 h, then proteins were transferred onto a nitrocellulose membrane (Hybond ECL, Cytiva) [18]. Reversible Ponceau Red staining (0.2% *w/v* Ponceau S in 3% *v/v* trichloroacetic acid) was applied in order to check equal protein loading. Afterward, the nitrocellulose membranes were washed 3 times in blotting solution (skimmed milk 3% *p*/*v*, triton X100 0.1% *v/v* in PBS pH 7.4) and, subsequently, incubated overnight at 4 °C with anti-APOA1 (sc376811-Santa Cruz Biotechnology, Dallas, TX, USA), anti-plasminogen (LF-MA0170- ThermoFicher, Waltham, MA, USA), anti-ceruloplasmin (LF-MA0159-ThermoFicher) diluted 1:1000. After 3 washing steps in blotting solution, membranes were incubated for 2 h with goat peroxidase-conjugated anti-mouse immunoglobulin G (Sigma, working dilution 1:3000). After 3 washing steps in blotting solution, a washing step in Triton X100 0.5% *v/v* in PBS pH 7.4 and 2 washing steps in 0.05 M Tris-HCl pH 6.8, the immunoreaction was performed using the ECL chemiluminescence detection system (Cytiva, Marlborough, MA, USA), and signals were detected by exposing membranes to Hiyperfilm ECL X-ray films (Cytiva, Marlborough, MA, USA). Wb images were analyzed by using the ImageMaster 2D Platinum v. 6.0 software (Cytiva, Uppsala, Sweden) using the relative integrated density values of detected spots. Immunoblot data was then exported into Excel (Microsoft Office, Redmond, WA, USA) to perform the Kruskal–Wallis two-tailed test statistical analysis and the post-hoc Dunn’s test for multiple comparison, using XLStat (Addinsoft, 2019, Microsoft Office, Redmond, WA, USA), version 2021.2.2.

### 2.7. Mass Spectrometry by MALDI-TOF

For protein identification, differentially abundant spots were manually exported from MS preparative gels and decolorized in 5 mM ammonium bicarbonate and 50% (*v/v*) acetonitrile solution, and then completely dehydrated in acetonitrile solution. Spots were digested in trypsin solution, overnight at 37 °C. Tryptic peptides purified by a C18ZipTip (Millipore, CA, USA), were rinsed with a 0.1% TFA and eluted directly on the MALDI target and dried. Then, 0.5 μL of saturated α-cyano-4-hydroxycinnamic acid (0.1% TFA and ACN) were added on dried spots and dried again. Peptides were then examined with an Autoflex™ Speed mass spectrometer (Bruker Daltonics, Bremen, Germany) armed with a Nd:YAG laser (355 nm; 1000 Hz) operated by FlexControl v3.3 and a nitrogen laser (355 nm). Spectra were acquired by delayed extraction technology with reflectron in positive mode. An average of 100 laser shots were used to improve the S/N ratio. Calibration was performed using a mixture of peptides containing angiotensin II 1046.54 *m/z*, bradykinin (fragment 1–7) 757.39 *m/z*, ACTH (fragment 18–39) 2465.19 *m/z*, renin substrate tetradecapeptide porcine 1760.02 *m/z*, and Glu fibrinopeptide B 1571.57 *m/z*. 

Protein identification was carried out using peptide mass fingerprinting (PMF) with the Mascot search engine, using SwissProt as database, homo sapiens as taxonomy, carbamidomethylation (Cys) and oxidation of methionine as fixed and variable modifications, respectively, one allowed missed cleavage, and a mass tolerance of 50 ppm. The MS data were deposited into the ProteomeXchange Consortium via the PRIDE [19] partner repository with the dataset identifier PXD029793.

### 2.8. T-SNE, Heat Map, and Correlation Analysis

For the multivariate analysis, the %V of differential protein spots variance in the overall cohort were used to perform t-Distributed Stochastic Neighbor Embedding (t-SNE) and a supervised heatmap analysis; the latter to visualize the abundance of the differential spots in each gel for the six conditions. Clustering of protein spots was performed on the basis of Euclidean distance. Another heatmap analysis limited to the differentially abundant protein spots corresponding to the time points for each treatment was also carried out. The above analyses and corresponding figures were obtained using RStudio Desktop 1.1.463 (Integrated Development for RStudio, Inc., Boston, MA, USA, https://www.rstudio.com; 11 October 2021) and XLStat (Addinsoft, 2019; Microsoft Office, Redmond, WA, USA). To determine the linear correlations of peripheral eosinophilia (eosinophil % of circulating granulocytes) and the %V of the proteins of interest, Pearson’s correlation coefficients were applied using software XLStat. 

### 2.9. Enrichment Analysis by MetaCore™ Software

The accession numbers of the proteins of interest, obtained from the comparison among the T0, T1, and T6 time points of each treatment follow-up (mepolizumab and benralizumab), were submitted to MetaCore network building tool software (Clarivate Analytics, Boston, MA, USA), separately. The MetaCore consists of a manually annotated database of protein interactions and metabolic reactions drawing from scientific literature. The uploaded data were processed using the shortest path algorithm set to high trust interactions. This algorithm allows to build a network including only closely related proteins and introducing a maximum of one non-experimental protein prioritized according to their statistical significance (*p* ≤ 0.001). Networks were graphically represented by nodes and edges, that correspond to proteins and the relationships between proteins, respectively. In order to compare the two different treatments, we also performed “Process Network” and “Pathway Maps” analyses, belonging to the MetaCore functional ontology enrichment tool which allows to analyze the uploaded data by considering Gene Ontology terms. These analyses provide a description of the biological functions or the signaling mechanisms in which the uploaded proteins are involved, respectively.

## 3. Results

### 3.1. Clinical Data

Table 1 and Table 2 report the characteristics and parameters of the patient cohort; in particular, Table 1 reports demographic data, ACT score, eosinophil count, serum IgE, and lung function parameters at T0 while Table 2 reports the same parameters in the time points. Particular differences in demographic and baseline functional and immunological data were not observed between groups. In proportion to time of therapy, all patients improved significantly in terms of asthma control, lung function parameters, and clinical status. In particular, FEV1 (in mL and %) and FEV1/FVC ratio significantly increased after 1 and 6 months of mepolizumab or benralizumab therapy associated with a significant decrease in peripheral eosinophilia (in cells/mm^3^ and %).

### 3.2. Proteomic Analysis

In our analysis, we compared proteomic profiles of serum from 9 patients affected by severe eosinophilic asthma before (T0), after 1 and 6 months of benralizumab (T1B and T6B; 4 patients) or mepolizumab (T1M and T6M; 5 patients) treatment and serum from control subjects (CTRL; 5 subjects). The comparison highlighted 82 differences between CTRL, T0, T1M, T6M, T1B, and T6B profiles, visible in Appendix A. The 58 identified protein spots, corresponding to 17 unique proteins, are reported in Appendix A. If we focus on benralizumab time points analysis, we highlighted 30 differences, while in the mepolizumab time points analysis there were 24 differences. 

### 3.3. T-SNE, Heatmap and Correlation Analysis

Relying on proteomic data, we performed a t-SNE analysis on all 82 differences. The t-SNE graph is reported in Figure 1, showing that samples that behave in the same condition group together well, corroborating that the differential protein pattern is characteristic for each condition. Moreover, T6 samples of both treatments cluster near the CTRLs. On the other hand, T6 samples together with CTRLs are localized on the opposite side with respect to T0 and T1 samples on the t-SNE 1 (Figure 1). T0 and T1 samples did not present considerable differences along t-SNE1, but they were evidently influenced by the two different treatments received, separating themselves along t-SNE2. On the contrary, T6 samples converged towards CTRL samples along t-SNE1, although they are still distinct by treatment. T-SNE analysis visibly grouped our samples in two principal groups, the A group, enclosed in red, includes T0, T1M, and T1B samples, and the B group, circled in green, includes T6M, T6B, and CTRL samples. 

### 3.4. Heatmap Analysis

Heatmap analysis (Figure 2) shows the spot abundance trend of significant matched spots in all gels; similar to t-SNE analysis, the general spots abundance trend separates analyzed subjects into two principal groups, A and B, as indicated by the dendrogram on the top of the matrix. Of note, the A group is composed by T0, T1M, and T1B samples with a distinct separation only by treatment, which shows an opposite pattern particularly evident in the center of the map (clusters C_3a and C_3b). The B group includes T6M, T6B, and control samples, presenting a similar protein profile to each other, but inverted with respect to A group, especially noticeable on the top of the heatmap (clusters C_4 and C_5). Nevertheless, in the lower section of the matrix (cluster C_1), T6 samples instead show a peculiar upregulation of protein species. Line charts at the right of the heatmap emphasize the trend of abundance of the grouped spots. 

### 3.5. Heatmap Analysis Performed on Significant Matched Spots and Pearson’s Correlation

Figure 3 shows the heatmap analyses relative to the proteins of interest for the two time points. In particular, Figure 3A reports the benralizumab time points and Figure 3B the mepolizumab ones. Both evidenced a reversed trend after 6 months of monoclonal treatment with respect to T0/T1. 

Therefore, we evaluated whether peripheral eosinophilia (eosinophil % of circulating granulocytes) were correlated with the %V of significant proteins detected by proteomic analysis, at T0 and T6 of each patient treated with benralizumab or mepolizumab. APOA1 and APOCs (APOC-II and APOC-III) results negatively correlated with peripheral eosinophilia (eosinophil % of circulating granulocytes) (Figure 4). 

### 3.6. MetaCore Analysis

The biological networks built by MetaCore shortest-path algorithm shows a complex interaction network of identified proteins for time points of the two different treatments (Figure 5). Alpha-2 macroglobulin (A2M), plasminogen, alpha-1-antichymotrypsin (SERPINA3), emerged as the functional hubs in both analyses; proteins with a higher number of interactions. Canonical paths referred to the networks are reported in Appendix A. The Pathway Maps and Process network comparisons of the differential proteins for time points of the two different treatments are presented in Figure 6. Among pathway maps, we found a series of terms related to HDL-mediated reverse cholesterol transport and lipoprotein metabolism, signaling pathways related to development, disruption of epithelial layer restitution in asthma, blood coagulation, and ECM remodeling, while reported among the process network are inflammation by the Kallikrein-kinin system and IL-6 signaling, iron transport, connective tissue degradation and ECM remodeling, cell adhesion by platelet–endothelium–leucocyte interactions, and again blood coagulation. 

### 3.7. Western Blot Validations

In order to visualize the trend of ceruloplasmin abundance after 6 months of treatment, we performed 1DEWB comparing the six conditions (T0, T1M, T1B, T6M, T6B, and CTRL). As shown in Figure 7, 1DEWB highlighted two bands of ceruloplasmin: the lower band shows an increased level already after 1 month of benralizumab administration as previously reported [13,14], remaining high also after 6 months of treatment, while in mepolizumab it increased after 6 months of treatment approaching the CTRL condition. The upper band increment is present only after 6 months of mepolizumab therapy. 

Since by proteomic analyses we found different protein species of plasminogen upregulated after 6 months of both treatments, we also validated the trend of abundance of plasminogen using two-dimensional Western blot analysis. Figure 8 shows, in accordance with proteomic analysis, a substantial increase in plasminogen signal, positively correlated with the follow-up of both therapies. 

Moreover, considering the proteomic results regarding APOA1, we validated this protein using 1DEWB as shown in Figure 9. APOA1 is characterized by a general upregulation of the protein after 6 months of both treatments. 

## 4. Discussion

Benralizumab and mepolizumab therapies are currently approved in the EU for SEA patients that report blood eosinophil count ≥300 cells/μL. Patients positively respond to treatments already after 1 month of administration, with an improvement in functional parameters, in particular with an increase in FEV1 and FEV1/FVC ratio and peripheral eosinophilia decrease in all patients. A study on benralizumab effects highlighted an improvement in mean FEV1 of 100 mL compared to placebo, but this effect was only seen in the higher eosinophil group [20], while the DREAM study on mepolizumab efficacy on exacerbation rate appears to be higher with greater blood eosinophil levels [21]. These results shed light on the importance of patient selection for a correct target therapy [20]. However, at present, if patients show a positive outcome the treatment should be continued indefinitely [10], making clear the necessity of new markers for prediction of treatment response. Although the detection of blood eosinophils is considered the best-established biomarker for efficacy of anti-IL-5 treatments [22], it does not provide a clear discrimination for IL-5-target drug choices and does not provide information about molecular changes at the peripheral level. In this regard, we applied a proteomic approach on serum of SEA patients, obtained before and after 1 and 6 months of mepolizumab or benralizumab treatment and compared to healthy controls. The analysis highlighted 82 differences between CTRL, T0, T1M, T6M, T1B, and T6B profiles and in particular, 30 referred to benralizumab time points (T0, T1B, T6B) and 24 to mepolizumab time points (T0, T1M, T6M) analysis. Once the t-SNE and heatmap analyses, were performed with the differential spots as observations, our data revealed that the differential serum protein profile at 6 months of both treatments is more similar to that of the healthy subjects. On the other hand, T1 serum protein profiles are more similar to that of T0 patients. 

When we consider singular time points analyses, the differential protein profile for benralizumab treatment shows that at T6B, the protein pattern behavior is opposite with respect to T1B/T0 and remains in part different from controls while the differential protein profile for mepolizumab treatment shows at T6M a behavior similar to controls and opposite to T1M and T0.

Enrichment analysis allowed to extrapolate data regarding process networks, and molecular pathways from our identified proteins in addition to building a protein network presuming the biological system where these proteins act. In particular, we performed an enrichment analysis on the two groups of differential proteins: the time points of benralizumab and the time point of mepolizumab, separately. Interestingly, we found that these two groups of differentially abundant proteins have similar pathway maps and process networks, in addition to a similar protein network that differs in benralizumab for a canonical path linking TTHY at A2M related through intra and extracellular L-Thyroxine, thyroid hormone receptor alpha, and myelin basic protein. Process networks associated to this path are: response to endogenous stimulus (75.0%; 3.563 × 10^21^), regulation of multicellular organismal process (82.5%; 1.279 × 10^19^), response to hormone (62.5%; 2.295 × 10^19^), response to nutrient levels (52.5%; 1.386 × 10^18^), and response to extracellular stimulus (52.5%; 3.908 × 10^18^).

Among enrichment analysis results, in both therapies, we highlighted a relevant involvement of lipid metabolism and transport. APOA1 and APOCs are the major High Density Lipoproteins (HDL) components found upregulated in both drugs in our patients. In particular, APOC-II was upregulated only in benralizumab. Interestingly, HDL apolipoproteins are reported to be responsible for anti-inflammatory and anti-allergic effects by suppressing neutrophil and eosinophil activation, adhesion, and chemotaxis [23]. Barochia et al. also reported that serum levels of HDL-components, in particular APOA1, are positively correlated with FEV1 in atopic asthma patients [24]. In agreement with this data, here, we also show for the first time that serum levels of APOA1, APOC-II, and C-III are negatively correlated with blood eosinophil count. APOA1 is also one of the functional hubs of both protein networks, further supporting its pivotal role.

Other interesting molecular pathways that we found using enrichment analysis are related to blood coagulation, plasminogen activator signaling, the role of alpha V/beta 6 integrin, regulation of IGF family activity, disruption of epithelial layer restitution in asthma, and the relationship with the upregulation of plasminogen (PLMN) that we found using proteomics after both treatments. PLMN is also a central functional hub in theoretical protein networks for both treatment time points, and it is related to inflammation, blood coagulation, proteolysis, regulation of angiogenesis, blood vessel morphogenesis, and platelet endothelium leucocyte interactions using process network analysis. It is reported that asthmatic patients are characterized by a prothrombotic state that increases with asthma severity [25]. Since after mepolizumab or benralizumab treatment we found a modulation of proteins involved in the thrombotic state, it might help to compare coagulation and fibrinolysis parameters during treatment follow-up. 

Alpha 2 macroglobulin (A2M) is reported, together with plasminogen, to be involved in the blood coagulation pathway. A2M is a cytokine transporter and protease inhibitor of trypsin, thrombin, and collagenase, already reported to be involved in severe asthma mechanisms such as regulation of airway remodeling [12,26]. Interestingly, A2M was downregulated after 6 months of mepolizumab or benralizumab and is considered a central hub of both networks focusing on its potential interesting role after MAbs treatments.

In our previous proteomic work on comparison of SEA serum protein profiles at T0 and T1 of both monoclonal therapies, we found upregulation of some protein species of ceruloplasmin (CERU) after 1 month of benralizumab treatment [12], assuming a lowering of asthma-induced oxidative stress. Our redox proteomic study [16] also confirmed a higher oxidation of ceruloplasmin after 1 month of both treatments, probably counteracting the asthma-induced oxidative stress. In the present study, considering a longer period of follow-up, we found the upregulation of the same protein species after 1 month of benralizumab, confirming our previous data, and interestingly, we validated an increase in the abundance of CERU also after 6 months of mepolizumab, approaching control sample levels, but not in benralizumab. 

In our above-mentioned work of redox proteomics, the results also highlighted that the protein species of TTHY were highly oxidized after 1 month of benralizumab with respect to mepolizumab, hypothesizing a potential role in regulation of oxidative stress [15]. In this 6-month follow-up study, TTHY is reported upregulated after benralizumab treatment and the protein network reports TTHY linked to APOC-III, CFB, SERPINA 3, and A2M via NF-kB transcriptional factor. SerpinA3, also called alpha 1 antichymotrypsin, another central hub in both networks is an acute phase protein that is upregulated after both monoclonals and is linked to cathepsin G, leukocyte elastase, kallikrein 3, granzyme B, and chymase in the protein networks. Indeed, the process network results refer these proteins to the extracellular matrix (ECM) remodeling [27]. 

All of these findings suggest that mepolizumab and benralizumab act on proteins involved in lipid metabolism and transport, blood coagulation, cell adhesion, and ECM remodeling. Although we found differently abundant proteins that are similar in function in the mepolizumab and benralizumab time points, we also reported some feature proteins for each follow-up. Our results suggest a monitoring of the parameters associated to the reported pathways during MAbs follow-up in order to further analyze molecular modulation at the peripheral level of mepolizumab and benralizumab therapies. 

Limitations: This represents a proteomic pilot study. Even if the number of analyzed samples in the 2-DE experiment is considered normal and adequate, the clinical record size is considered quite low. For these reasons, all of the differentially abundant proteins will be further analyzed in a larger cohort using a different approach such an ELISA, WB test, and/or targeted proteomic techniques.

## 5. Conclusions

Asthma represents a diffused chronic disease that is rapidly increasing worldwide, and asthma attacks can be scary and impair quality of life. Treatments lead to symptom relief, especially in mild asthma. However, for severe asthma new biological therapies have been introduced to inhibit molecular pathways leading to severe symptoms; in particular, mepolizumab and benralizumab act against the IL-5 pathway. Different studies reported that already after 1 month of both treatments, patients reported a significant symptomatic improvement in terms of functional and immunological parameters, as well as clinical status. In addition to these parameters, there is the urgency to clarify the follow-up of both therapies to identify biomarkers for the optimal duration of treatment, persistence of effectiveness, and safety. 

This work aimed to highlight the molecular changes at the serum level after mepolizumab and benralizumab at different time points of follow-up: 1 and 6 months of both treatments. Our results show that after 6 months of mepolizumab and benralizumab, the serum protein profile follows that of controls, in particular for mepolizumab. Modulation in protein abundance is particularly observed for proteins involved in inflammatory pathways, regulation of blood coagulation, lipid metabolism, and ECM remodeling as suggested by enrichment analysis. In particular, we detected upregulation of certain protein species of APOA1, APOC-II, and APOC-III, which resulted as negatively correlated with peripheral eosinophilia (cell/mm^3^%) and upregulation of certain protein species of the antioxidants ceruloplasmin and transthyretin. Interestingly, a particular behavior was detected for plasminogen, which is lower in CTRL and T0 groups and tends to increase simultaneously with time points of both therapies. These data, further confirmed in a wider cohort of patients, could be evaluated at clinical level in order to monitor patients during biological therapies for the optimal duration of treatment, persistence of effectiveness, and safety.

## Figures and Tables

**Figure 1 biomedicines-10-00761-f001:**
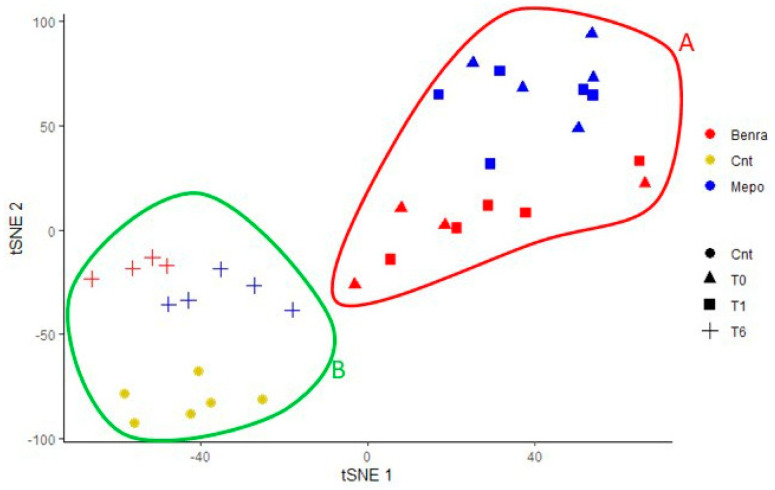
T-SNE plots of the six conditions were created using the %V data of all significant spots according to the variance in the overall cohort. Red circle includes T0, T1M, and T1B samples. Green circle includes T6M, T6B, and CTRL samples.

**Figure 2 biomedicines-10-00761-f002:**
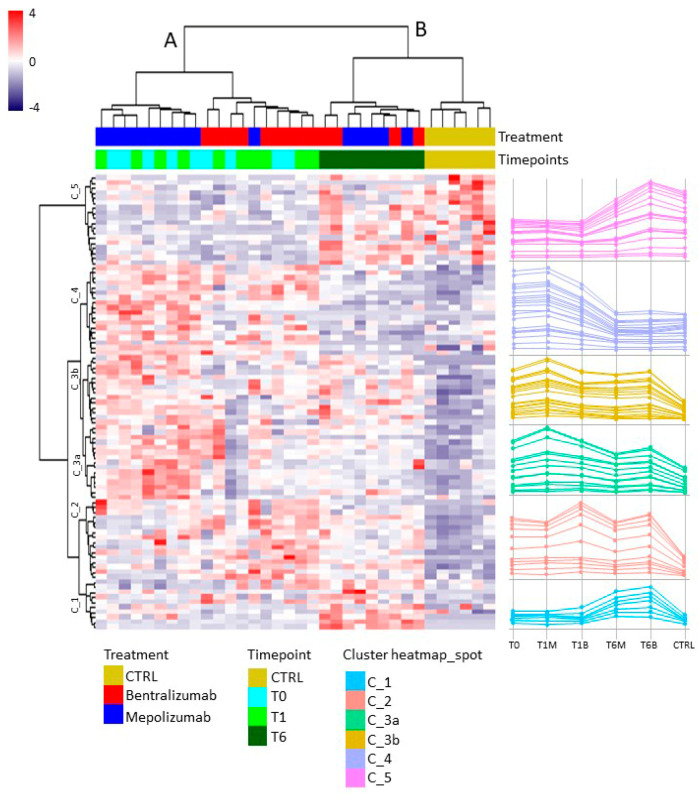
Heatmap analysis performed on significant matched spots in all gels among the six conditions (T0-T1B-T1M-T6M-T6B-CTRL). Blue to red colors in the heatmap graph correspond to protein abundance, as reported by the legend. On the right, the line charts show the trend of abundance of the significant clustered spots.

**Figure 3 biomedicines-10-00761-f003:**
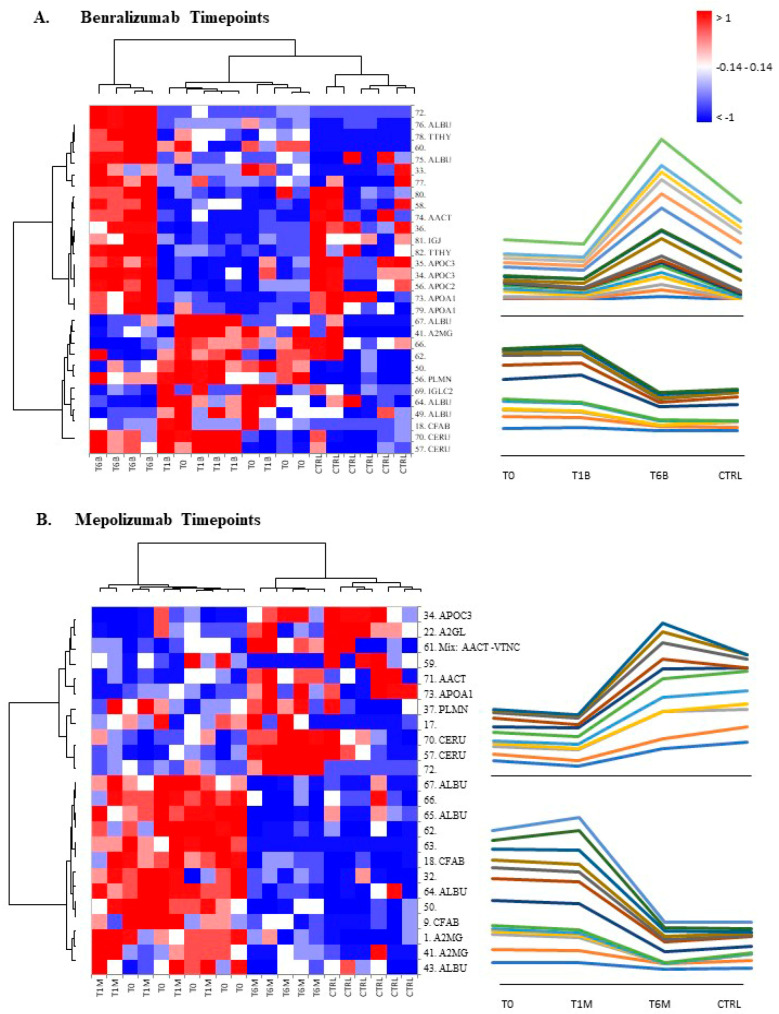
Heatmap analysis performed on significant matched spots in benralizumab time points analysis (**A**) and mepolizumab time points analysis (**B**). On the right, the line charts show the trend of abundance of the clustered spots.

**Figure 4 biomedicines-10-00761-f004:**
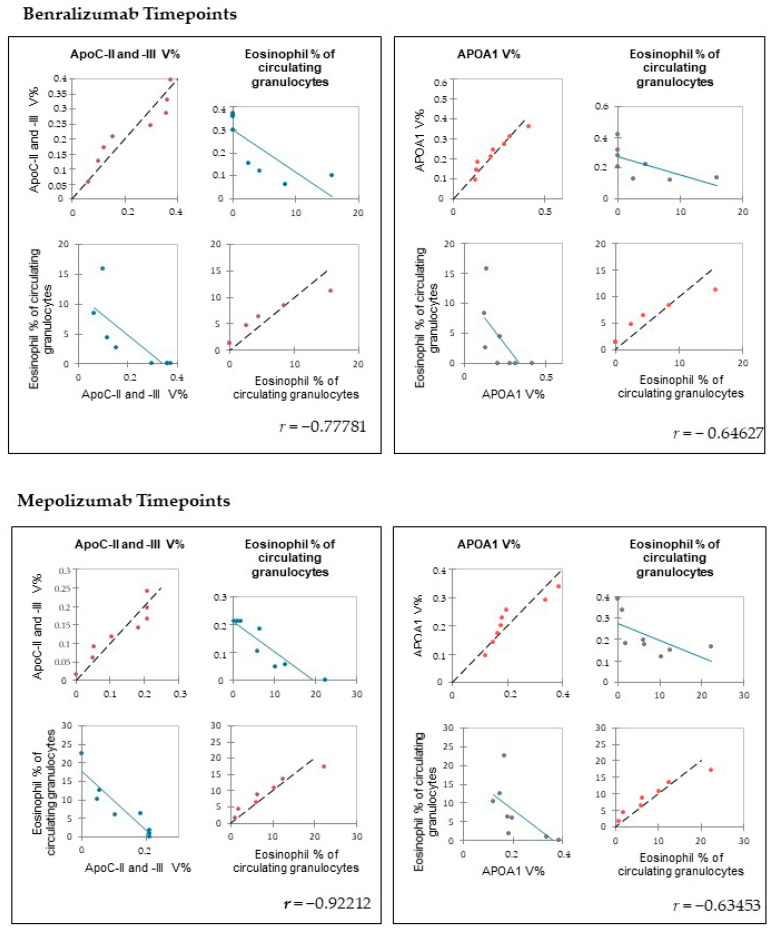
Pearson’s correlation between eosinophil % of circulating granulocytes in SEA patients and APOA1, APOC-II, and -III %V in serum at T0 and T6 time points. Values are shown with Pearson’s correlation coefficients^®^.

**Figure 5 biomedicines-10-00761-f005:**
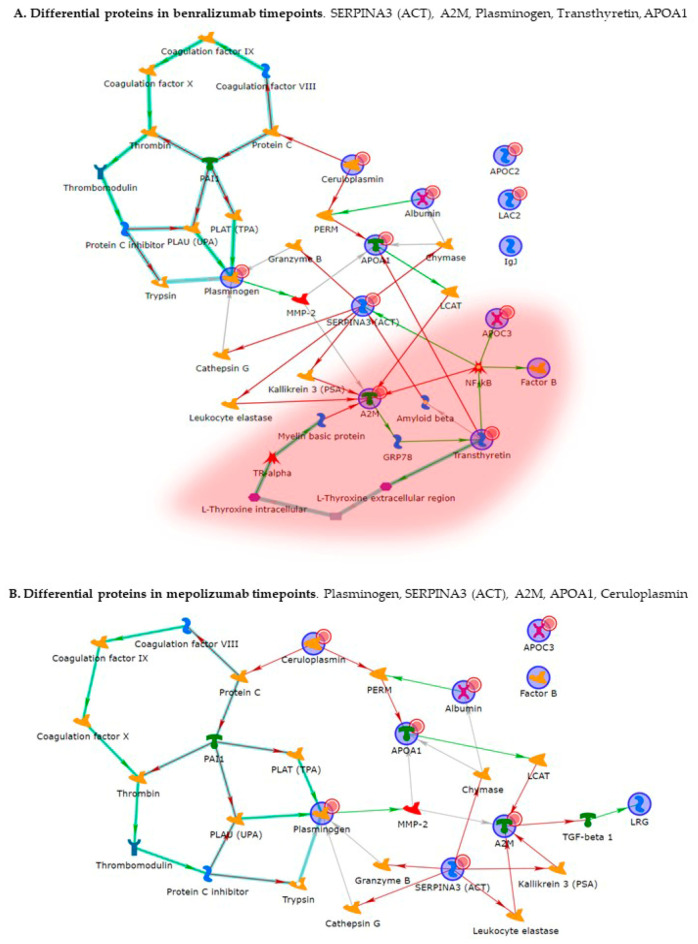
MetaCore analysis. (**A**) Network related to the differential proteins in benralizumab time points (T0, T1B, T6B). (**B**) Network related to the proteins in mepolizumab time points (T0, T1M, T6M). Red circle indicates the canonical pathway involving transthyretin, which was found differential only in benralizumab time points analysis. Statistical significance of protein interaction was *p* ≤ 0.001.

**Figure 6 biomedicines-10-00761-f006:**
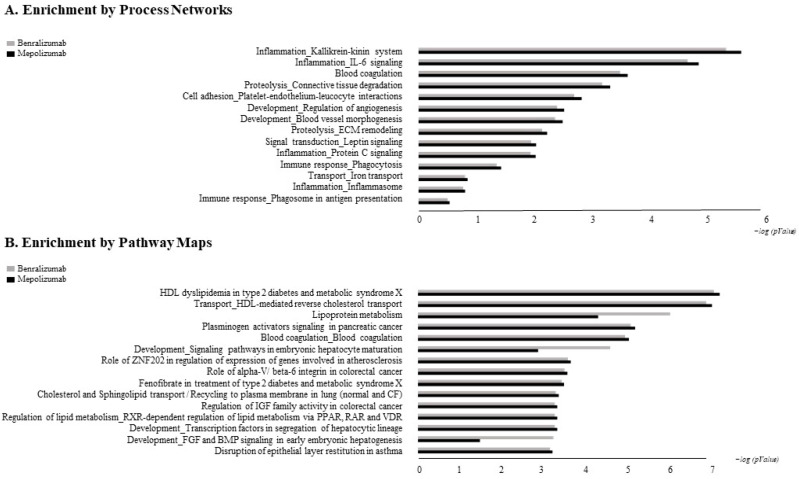
MetaCore analysis: comparison of biological processes (**A**) and pathway maps (**B**) by gene ontology differential proteins in benralizumab and mepolizumab time points. Black histograms show mepolizumab time points; grey histograms show benralizumab time points.

**Figure 7 biomedicines-10-00761-f007:**
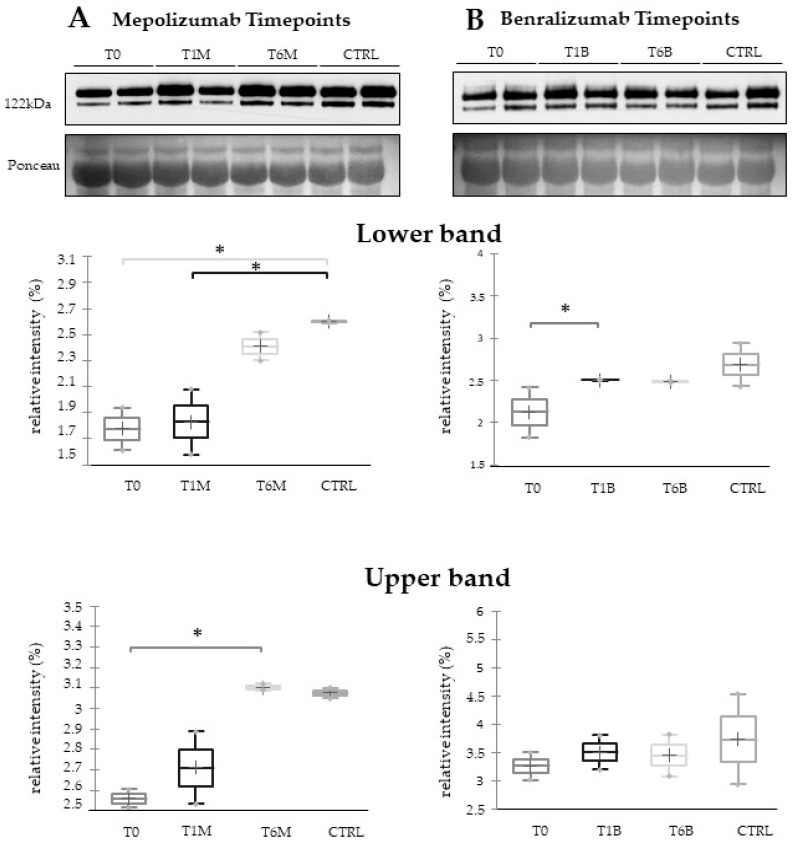
The 1D-Western blot analysis with anti-ceruloplasmin monoclonal antibody on serum samples from SEA patients at T0, T1M, and T6M (**A**) or T0, T1B, and T6B (**B**) and CTRLs. Kruskal–Wallis statistical analysis validates the lower band upregulation of ceruloplasmin in T1B serum and the upper band upregulation of ceruloplasmin in T6M serum. * Indicates significant level *p* < 0.05.

**Figure 8 biomedicines-10-00761-f008:**
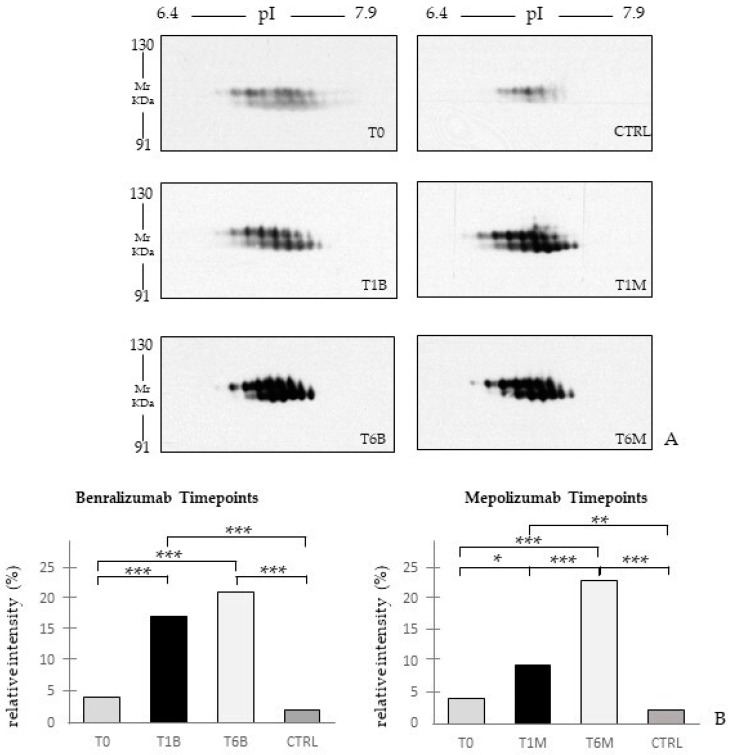
(**A**) The 2D-Western blot analysis with anti-plasminogen antibody on serum sample SEA patients at T0, T1M, T6M, T1B, and T6B, CTRL. (**B**) Histograms report the relative intensities of the spots and the HSD Tukey *p*-values. * Indicates significant level *p* < 0.05. ** Indicates significant level *p* < 0.005. *** Indicates significant level *p* < 0.0001.

**Figure 9 biomedicines-10-00761-f009:**
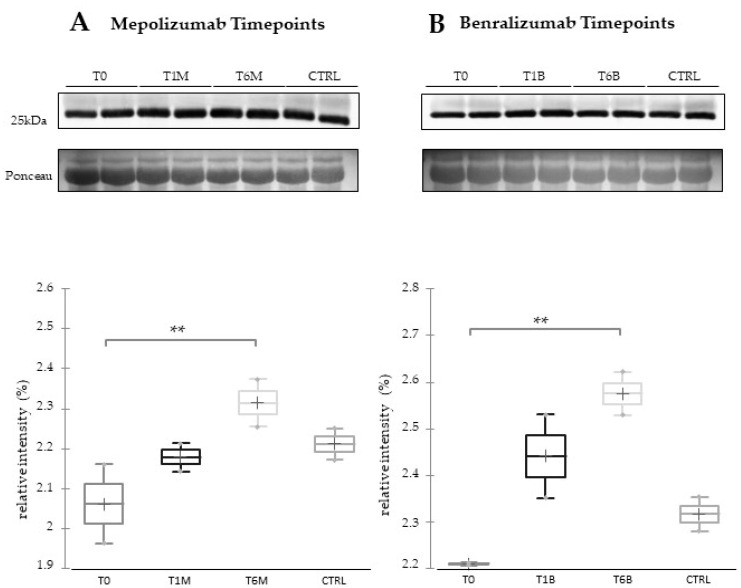
The 1D-Western blot analysis with anti-APOA1 antibody on serum samples of SEA patients at T0, T1M, and T6M (**A**) or T0, T1B, and T6B (**B**) and CTRLs. Kruskal–Wallis statistical analysis validates the upregulation of APOA1 after 6 months of monoclonal therapies (** = *p* ≤ 0.025).

**Table 1 biomedicines-10-00761-t001:** Demographic, clinical, immunological, and functional parameters of study population at baseline.

Baseline Characteristics	Mepolizumab-Treated Patients (*n* = 10)	Benralizumab-Treated Patients (*n* = 8)	Healthy Controls (*n* = 7)	*p* Values
**Age**	56 ± 11	50.5 ± 16.1	45.3 ± 15.7	ns
**Gender (F/M)**	2/8	4/4	5/3	ns
**Smoking Habits** **(former–current/never)**	6/4	6/2	2/5	ns
**Age at onset**	33.2 ± 16.6	34.3 ± 17	n.a.	ns
**OCS dosage**	3.7 ± 4.3	4 ± 4.5	n.a.	ns
**Eosinophil cell count (cell/mm^3^)**	1090 ± 821	719 ± 419	n.a.	ns
**Eosinophil cell count (%)**	11.6 ± 7	8.2 ± 4.6	n.a.	ns
**Functional parameters:**				
**FEV1(%)**	77 ± 27	81 ± 25	n.a.	ns
**FVC**	92 ± 21	99 ± 23		ns
**TIFFENAU index**	66 ± 16	66 ±8		ns
**ACT Score**	14 ± 6	14.8 ± 6.8	n.a.	ns

Table repots oral corticosteroid dosage expressed as prednisone equivalent. OCS: oral corticosteroids; ACT: asthma control test; ACQ: asthma control questionnaire. (n.a = not applicable, ns = not significant).

**Table 2 biomedicines-10-00761-t002:** Comparison of clinical and functional parameters.

	Mepolizumab Group				Benralizumab Group			
	T0M	T1M	T6M	*p* Values	T0B	T1B	T6B	*p* Values
**ACT Score**	14 ± 6	20 ± 5	21.8 ± 3.5	0.005	14.8 ± 6.8	21 ± 3	23.8 ± 0.9	0.004
**Functional parameters:**								
**FEV1 (%)**	77 ± 27	88 ± 22	86 ± 22	0.03	81 ± 25	89.7 ± 15	98 ± 15	0.04
**FVC (%)**	92 ± 21	96 ± 21	94 ± 18	ns	94 ± 23	96 ± 21	107 ± 18	ns
**FEV1/FVC**	66 ± 16	72 ± 9	72.7 ± 10	0.02	66 ± 8	71 ± 9	73 ± 10	0.04
**Peripheral eosinophilia (cell/mm^3^/%)**	1090 ± 821	160 ± 108	167 ± 115	<0.0001	718 ± 49	0	0	<0.0001
**Peripheral eosinophilia**	11.6 ± 7	1.9 ± 1.5	2.2 ± 1.4	<0.0001	8.2 ± 4.6	0	0	<0.0001

Table reports eosinophil cell count and FEV1% in the time points of mepolizumab and benralizumab groups. (ns = not significant).

## Data Availability

The mass spectrometry proteomics data have been deposited to the ProteomeXchange Consortium via the PRIDE partner repository with the dataset identifier PXD029793.

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
