# Peer review of "Serum Proteomic Profile of Asthmatic Patients after Six Months of Benralizumab and Mepolizumab Treatment"

_biomedicines, 2022, doi:10.3390/biomedicines10040761_

Round 1
Reviewer 1 Report
The article is well-written and focuses on an interesting topic. Materials and methods have been extensively described and the discussion motivates the reader in further research. However, the conclusions appear to be quite weak in comparison to the other sections: how do the results presented by the authors affect the clinical practice of physicians? This remains quite unclear.
Author Response
Point 1: The article is well-written and focuses on an interesting topic. Materials and methods have been extensively described and the discussion motivates the reader in further research. However, the conclusions appear to be quite weak in comparison to the other sections: how do the results presented by the authors affect the clinical practice of physicians? This remains quite unclear.
We thank the Reviewer for this comment. At present, the SEA patients that show a positive outcome after monoclonal treatments should continue the therapy indefinitely, however, there are not markers for prediction of treatment response. Our results, furtherly confirmed in a wider cohort of patients, could be evaluated at clinical level in order to monitor patients during the biological therapies for the optimal duration of treatment, persistence of effectiveness and safety. Moreover, there is not a clear parameter used to discriminate for IL-5-target drugs choice, along this line we also reported some feature proteins involved in specific processes for each follow-up.
We changed the conclusion according to the Reviewer suggestion (Pages 18-19, lines 484-509).
Pages 18-19, lines 484-509
“Asthma represents a diffused chronic disease in rapid increase around the world and asthma attacks can be scary and impair the quality of life. Treatments lead to relieve the symptoms especially in mild asthma but for severe asthma have been introduced new bi-ological therapies to inhibit molecular pathways leading to severe symptoms, in particular, mepolizumab and benralizumab act against IL-5 pathway. Different studies reported that already after one month of both treatments, patients referred a significant symptomatic improvement in terms of functional and immunological parameters, as well as clinical status. In addition to these parameters, there is the urgency to clarify the follow-up of both therapies to identify biomarkers for the optimal duration of treatment, persistence of effectiveness and safety.
This work aims to highlight the molecular changes at the serum level after mepolizumab and benralizumab in different time point of follow-up: 1 month and 6 months of both treatments. Our results show that after six months of mepolizumab and benrali-zumab, serum protein profile tends to that of controls, in particular for mepolizumab. Modulation in protein abundance is particularly observed for proteins involved in inflammatory pathways, regulation of blood coagulation, lipid metabolism and ECM re-modeling as suggested by enrichment analysis. In particular, we detected up-regulation of certain protein species of APOA1, APOC-II and APOC-III, which resulted negatively correlated with peripheral eosinophilia (cell/mm3%) and up-regulation of certain protein species of the antioxidant ceruloplasmin and transthyretin. Interestingly, a particular behavior is detected for plasminogen, which is lower in CTRL and T0 groups and tend to increase simultaneously with time point of both therapies. These data, furtherly confirmed in a wider cohort of patients, could be evaluated at clinical level in order to monitor patients during the biological therapies for the optimal duration of treatment, persistence of effectiveness and safety..”
Reviewer 2 Report
Dear Auhtoes I have read the manuscript and I send you my comments:
1I Methods: please add data on power calculation
Author Response
Dear Auhtoes I have read the manuscript and I send you my comments:
Point.1 Methods: please add data on power calculation
A1. We thank the Reviewer for this comment. Because proteomic techniques are costly and time-consuming, studying large numbers of patients is not possible. In this setting, due to our dataset accounted thousand variables, we mainly focused on avoiding type 1 errors through the postHoc correction. However, we opted for the Benjamini-Hochberg FDR rather than other corrections (like Bonferroni) because BH-FDR resulted in lower rate of the type 2 errors. We did not perform the power analysis before the data collection due to the patient availability was a limiting aspect. Albeit we might assume a higher Cohen effect size considering the overall high correlation of our variables, the power value of our analysis would be not appropriate; considering a sample size of 5 pairs, a significant level of 0.05 and a Cohen effect size of 0.75, the power would be close to 0.25. The recommended sample size to have a decent power (power > 0.8) would require 32 cases (16 pairs) but we could not have the opportunity to recruit this number of patients.
The size of this population is not considered too small for a preliminary proteomic study since proteomic is time and cost consuming. Moreover, the incidence of severe asthma patients that are treated with biologicals is considered not so frequent. For these reasons this population of samples could be considered appropriate for a preliminary proteomic analysis.
Reviewer 3 Report
dear Authors,
I have read the manuscript and I send you my comments:
1) Please indicate if it is a pilot study, add also the power calculation
2) Experimental protocol must be clarified.
Author Response
I have read the manuscript and I send you my comments:
Point 1. Please indicate if it is a pilot study, add also the power calculation
A1. We thank the Reviewer for this comment. Our study is a pilot study. Because proteomic techniques are costly and time-consuming, studying large numbers of patients is not possible. In this setting, due to our dataset accounted thousand variables, we mainly focused on avoiding type 1 errors through the postHoc correction. However, we opted for the Benjamini-Hochberg FDR rather than other corrections (like Bonferroni) because BH-FDR resulted in lower rate of the type 2 errors. We did not perform the power analysis before the data collection due to the patient availability was a limiting aspect. Albeit we might assume a higher Cohen effect size considering the overall high correlation of our variables, the power value of our analysis would be not appropriate; considering a sample size of 5 pairs, a significant level of 0.05 and a Cohen effect size of 0.75, the power would be close to 0.25. The recommended sample size to have a decent power (power > 0.8) would require 32 cases (16 pairs) but we could not have the opportunity to recruit this number of patients. We added “Limitation” paragraph in order to classificate of the pilot study (page 18, lines 479-483)
Page 18, lines 479-483
“Limitations: This represents a proteomic pilot study. Even if the number of analyzed samples in 2-DE experiment is considered normal and adequate, the clinical record size is considered quite low. For these reasons, all the differentially abundant proteins will be further analysed in a larger cohort using different approach such an ELISA, WB test and/or targeted proteomic techniques.”
Point 2. Experimental protocol must be clarified.
A2. We thank the Reviewer for the suggestion, we added a brief paragraph to clarify the Experimental design (page 3, lines 87-106), we also added a graphical abstract aimed at clarifying experimental protocol and results.
Page 3, lines 87-106
“Experimental design
Eighteen patients with severe eosinophilic asthma that reported blood eosinophil count ≥300 cells/μl were treated with monoclonal therapies. Ten patients were treated with mepolizumab and eight with benralizumab. The serum samples were collected at three different times: pre-treatment (T0), after 1 month of treatment with benralizumab (T1B) or mepolizumab (T1M) and after 6 months of treatment with benralizumab (T6B) or mepolizumab (T6M). The cohort of patients was in part used for proteomic analysis and in part used for validations by WB. Samples used for proteomic analysis were randomly selected.
Differential proteomic analysis by two-dimensional electrophoresis was performed on 5 patients treated with mepolizumab and 4 patients treated with benralizumab at different timepoints taking into consideration also healthy controls. Spots that resulted statistically significant from the comparison were identified by mass spectrometry MAL-DI-TOF. Identified proteins were submitted to enrichment analysis in order to highlight protein position into specific molecular pathways. Proteins with central functional roles and their behavior were validated by mono and bidimensional western blot in an alternative cohort of samples to confirm their particular pattern and their amount in precise time point of benralizumab and mepolizumab treatments. Proteomic data of the differential proteins were also correlated with peripheral eosinophilia (cell/mm3%) by Pearson’s linear correlation..”
Reviewer 4 Report
The paper presents serum profiles in severe eosinophilic asthma patients one month and 6 months after treatment with humanized anti –IL5 antibodies. The presented topic is very interesting and important in the research of biomarkers to identify the success, persistence and optimal duration for the treatment with biologicals.
General comments
This article could benefit from some revision in English, as some phrases sound a bit strange (e.g. lines 48-50, 351-353), there are some typos (line 300, 319).
The authors state that 18 patients with SEA were enrolled in the study, but for proteomic analysis, they use only half of the patients (4 with benralizumab and 5 with mepolizumab). Based on what criteria did they choose those patients? Also, they use 5 control subjects (CTRL), but no data about their health status is offered. Control subjects should be included in Tabel 1.
What exactly is shown in Figure 4 charts? To which timepoints the correlation (r-value) is referring to?
Minor comments
Can the authors please specify how they measured the amount of protein (lines 104-106)?
Line 152 is „blotting buffer” or blocking buffer? lines 155-157, do you mean washing steps?
Could the authors add a small section with the abbreviations used throughout the article?
Can the authors please clarify what they mean by process network versus molecular pathway?
The authors should label the y axis in figures 7,8,9. Also, I don’t think is necessary to add both „*” and p-value, maybe use the symbol in the figure and the actual value in the text.
Reviewer 5 Report
First of all, I would like to congratulate the authors for a well thought out and developed work.
However, I would like the authors to make some changes/clarifications that I believe would improve the presentation of the paper:
1.- Regarding the spelling of the names Benralizumab and Mepolizumab, the authors should decide whether to write them in upper or lower case and maintain this decision throughout the text (see title as an example).
2.- In the last sentence of the introduction, as well as elsewhere in the article including the conclusion, the authors indicate that they include healthy controls. However, the demographic data (Table 1) and functional parameters (Table 2) do not provide any information on the data of the controls (how many control individuals?, how many on each of the techniques/experiments? ). Authors should include these data and justify their response.
3.- In the MetaCore analysis, it is not clear wich time points were taken to perform the comparison. Authors also should reflect the p value the authos has stablished to set the threshold of routes shown.
4.- Figures 5 and 6 should be improved, there is no A and B in the figure, but present in the legend. Also, in figure 6A there is no title.
5.- Monodimensional and Two-Dimensional Western blot where performed only from four serum samples of each condition. In the proteomic analysis, 4 patients from the benralizumab and 5 from the mepolizumab group were selected. How were this patients selected from each group?, are the same patientes in both analysis?. This confusion about the number of patients and controls in each of the analyses/techniques should be clarified. Why were not all individuals used in all techniques?, how were they selected for each section?
6.- In figure 7 and 9, how the "Kruskal-Wallis statistical analysis" was performed?. Please give more info about this statistical analysis.
